# The Sperm DNA Fragmentation Assay with SDF Level Less Than 15% Provides a Useful Prediction for Clinical Pregnancy and Live Birth for Women Aged under 40 Years

**DOI:** 10.3390/jpm13071079

**Published:** 2023-06-29

**Authors:** Shiao Chuan Chua, Steven John Yovich, Peter Michael Hinchliffe, John Lui Yovich

**Affiliations:** 1PIVET Medical Centre, Perth, WA 6007, Australia; schua@pivet.com.au (S.C.C.);; 2Hospital Shah Alam, Shah Alam 40000, Malaysia; 3Curtin Medical School, Curtin University, Perth, WA 6102, Australia

**Keywords:** sperm DNA fragmentation (SDF), Halosperm test, assisted reproductive technology (ART), in vitro fertilization (IVF), intracytoplasmic sperm injection (ICSI), IVF-ICSI Split, female age effects

## Abstract

This retrospective cohort study was conducted on 1148 males who presented along with their partners for infertility management at the PIVET Medical Centre between 2013 and 2022 and had a sperm DNA fragmentation (SDF) assay performed by Halosperm, thereafter participating in 1600 assisted reproductive technology (ART) cycles utilising one of three modalities, namely, IVF-Only, ICSI-Only or IVF–ICSI Split cycles. The outcomes from the ART cycles were then analysed as two groups based on SDF levels <15% and ≥15%. The study showed the unadjusted fertilization rates were not different between the groups, neither across the four female age ranges. However, when the fertilization rates were adjusted for the mature oocytes (metaphase-II oocytes), there was a highly significant difference in fertilization rates in favour of the group with SDF levels < 15% where the women were in the younger age grouping of <35 years (78.4% vs. 73.0%; *p* < 0.0001). Overall, there was no difference in the rates of blastocyst development nor clinical pregnancy rates between the two SDF groups, but there was a significantly higher pregnancy rate for the younger women (<35 years) with the group of SDF level < 15% (44.1% vs. 37.4%; *p* = 0.04). Similarly, there was no difference in the miscarriage rates overall with respect to SDF groups, and no clear picture could be deciphered among the women’s age groups. With respect to cumulative live births, this reflected the pregnancy rates with no overall difference between the two SDF groups, but there was a significantly higher cumulative live birth rate for women <35 years where the SDF level was <15% (38.6% vs. 28.6%; *p* < 0.01). Among the three modalities, the highest cumulative live birth rate occurred within the group with SDF level < 15%, being highest with the IVF mode, particularly for women aged <40 years (43.0% vs. 37.7% for IVF-ICSI Split and 27.9% for ICSI; *p* = 0.0002), noting that the IVF case numbers were disproportionately low.

## 1. Introduction

Male infertility can be caused by a range of male factors, including abnormal semen parameters or function, abnormalities of the reproductive system (anatomical, endocrine, genetic, functional, or immunological), acute or chronic illnesses, and sexual conditions that are incompatible with the capacity to deposit semen in the vagina [1]. The World Health Organization (WHO) defines male factor infertility as the presence of ≥1 abnormalities in the semen analysis or the presence of inadequate sexual or ejaculatory function [2]. Of all infertility cases, nearly 50% of cases are caused by the male factor, either as a single factor or in conjunction with the female factor [3,4]. Approximately 15% of infertile men were found not only to have normal seminal profiles [5], but also with sperm DNA fragmentation (SDF) of 15% and more [6]. 

Semen analysis remains the cornerstone for male infertility assessment [7] but has limitations. Albeit it provides useful information for the initial evaluation of an infertile man, apart from azoospermia, none of the semen measures, alone or in combination, is diagnostic of infertility [8,9]. Normal sperm analysis does not ensure the fertilization capacity of sperm, and studies have revealed significant overlap between fertile and infertile males in terms of sperm parameter values [10]. It is therefore imperative to acknowledge the limitations of semen analysis results in predicting the health and functional capacity of the male reproductive organs and cells. Hence, technological advancements in the field of andrology are required to produce reliable and cost-effective useful sperm function tests for the evaluation of infertile males [11].

WHO regards sperm DNA fragmentation (SDF) assay as a promising tool in the evaluation of sperm DNA integrity and function, and regards it as being predictive of assisted reproductive technology (ART) outcomes, which is in keeping with many studies [12,13,14,15,16,17]. The European Society of Human Reproduction and Embryology position report also indicates that impaired sperm DNA integrity may have negative effects on ART clinical outcomes, but the society urged high-quality clinical data are required in the evaluation of these outcomes [18]. On the other hand, the Practice Committee of the American Society for Reproductive Medicine (ASRM) does not recommend routine SDF analysis in the initial evaluation of the infertile couple, and reported that despite the multitude of systematic reviews attempting to decipher the effect of sperm DNA integrity on ART outcomes, heterogeneity has precluded any definitive findings about the clinical value of sperm DNA fragmentation prior to ART treatments [19,20]. Notwithstanding this viewpoint, ASRM recommended SDF analysis in evaluation of couples with recurrent pregnancy loss. The controversial data on clinical relevance of SDF in ART has been leading to a diversity of clinical practice recommendations over more recent decades.

There were five meta-analyses reported in 2008, 2012, 2014, 2017 and 2019, respectively, which concluded a direct association between sperm DNA damage and pregnancy loss after IVF/ICSI [15,21,22], increased SDF associated with lower pregnancy rates in IVF but not in ICSI cycles [22], lower clinical pregnancy rates after IVF/ICSI [23] and increased idiopathic recurrent pregnancy loss [24]. A more recent publication in 2021 which provides the largest evidence to date, undertook a meta-regression analysis where 67 studies were accepted from an electronic search of 859 records involving four different methods for detecting SDF, i.e., TdT (terminal deoxynucleotidyl transferase)-mediated dUDP nick-end labelling (TUNEL), sperm chromatin structure assay (SCSA), the alkaline comet assay and the sperm chromatin dispersion (SCD) test, again concluding that there was a negative association between sperm DNA damage and IVF outcomes. The study showed significantly reduced implantation, pregnancy rates and live birth rates from IVF but not from ICSI [25]. In contrast, a meta-analysis study undertaken in 2016 comprising 12,380 IVF/ICSI cycles from a systematic review of 25,639 IVF/ICSI cycles indicated that pregnancy rates were negatively influenced by sperm DNA damage but, after adjustment for publication bias, no conclusion could be drawn regarding the relevance on pregnancy rates [26]. Against this background, our study was designed with the primary aim to assess the impact of specific SDF levels namely <15% and ≥15% on the laboratory and clinical outcomes within ART treatments; and a secondary aim was to evaluate those laboratory and clinical outcomes among three ART subgroups, namely, IVF–ICSI Split, IVF-Only and ICSI-Only cycles. 

## 2. Materials and Methods

### 2.1. Study Design

This retrospective cohort study was conducted on 1148 males involved within 1600 ART cycles who presented along with their partners for infertility management at the PIVET Medical Centre between 1 March 2013 and 1 March 2022 and had an SDF test. Inclusion criteria were males with an SDF assay performed by Halosperm who underwent conventional IVF-Only or ICSI-Only or an IVF–ICSI Split cycle with their own fresh or frozen ejaculated spermatozoa using their partner’s autologous or recipient oocytes. We routinely performed an SDF assay in PIVET unless it was declined by the patient. Patient clinical data including demographics, cycle characteristics, laboratory and clinical outcomes were extracted from a FileMaker Pro 12 database management system. 

Two groups were created for comparisons using 15% DNA fragmentation index (DFI) as the cut-off point based on published research which found a significantly lower rate for failure of fertilization [27] and higher blastulation rate [28] at DFI < 15%. (Both DFI and SDF are used interchangeably in this manuscript and its associated literature).

The SDF < 15% arm has 862 participants and the SDF ≥ 15% arm has 286 participants. All participants were also sub-categorized based on the three ART modalities. From Figure 1 it can be seen that 76 cycles of IVF-Only were initiated on 71 couples, 1295 cycles of ICSI-Only were initiated on 860 couples whilst the IVF-Split Modality was initiated on 229 cycles comprising 217 couples during the defined study period. 

### 2.2. Definitions of Clinical and Laboratory Outcomes

In this study, we refer clinical and laboratory definitions from a consensus and an evidence-driven set of terms and definitions by The International Glossary on Infertility and Fertility Care [1], led by The International Committee for Monitoring Assisted Reproductive Technologies (ICMART) in partnership with several substantial ART societies, the Vienna consensus report on the development of ART laboratory performance indicators [29], and National Perinatal Epidemiology and Statistics Unit (NPESU) [30] definition as specified below. Our main laboratory outcomes include rates of fertilization, overall blastocyst development and good-quality blastocyst, whilst clinical outcomes were clinical pregnancy rate, miscarriage rate and live birth rate. 

#### 2.2.1. Mature Oocyte

An oocyte in metaphase II of meiosis (MII) that displays the first polar body and is capable of fertilization.

#### 2.2.2. Fertilization (Normal)

A sequence of biological processes initiated by the entry of a spermatozoa into a mature oocyte followed by creation of two pronuclei (2PN). 

#### 2.2.3. Intracytoplasmic Sperm Injection (ICSI)

A procedure in which a single spermatozoon is injected into the cytoplasm of an oocyte. ICSI normal fertilization rate was defined as the number of oocytes containing 2PNs per MII oocytes injected, assessed at 17 ± 1 h post injection (Day 1).

#### 2.2.4. In Vitro Fertilization (IVF)

A sequence of procedures involving the extracorporeal fertilization of gametes. IVF normal (unadjusted) fertilization rate was defined as the number of fertilized oocytes on Day 1 per number of Cumulus-Oocyte Complexes (COCs) inseminated. Due to the presence of undisclosed immature or defective oocytes at insemination, the normal, unadjusted IVF and ICSI group fertilization rates are not entirely comparable. We therefore also recorded an adjusted IVF fertilization rate as 2PNs per mature (MII) oocytes for a fairer statistical comparison.

#### 2.2.5. Blastocyst

Blastocyst describes the stage of preimplantation embryo development that occurs approximately 5–6 days following insemination or ICSI and is categorized in accordance with Gardner’s blastocyst grading system. The blastocyst contains blastocoele cavity, trophectoderm and an inner cell mass. Blastocyst rate was calculated as the total number of blastocysts per total number of 2PNs occurring. A good-quality blastocyst is categorized with Gardner scoring of 3BB and greater with the rate expressed per total 2PNs.

#### 2.2.6. Embryo Transfer (ET)

ET denotes placement of an embryo at any embryonic stage from Day 1 to Day 7 after IVF or ICSI into the uterus. At PIVET, currently ~98% are single embryo transfer (SETs).

#### 2.2.7. Frozen-Thawed Embryo Transfer (FET)

FET denotes an ART procedure in which cycle monitoring is carried out with the intention of transferring a frozen/thawed (actually vitrified/warmed) embryo(s)/blastocyst(s) to a woman.

#### 2.2.8. Clinical Pregnancy

A pregnancy diagnosed by ultrasonographic detection of one or more gestational sacs or definitive clinical signs of pregnancy. It includes a clinically documented ectopic pregnancy (by ultrasound or surgical diagnosis). Clinical pregnancy rate was calculated as the number of clinical pregnancies per 100 embryo transfer cycles.

#### 2.2.9. Miscarriage

Spontaneous loss of a clinical pregnancy in which the embryo(s) or fetus(es) is/are nonviable before a gestational age of 20 weeks [30]. Miscarriage rate was calculated as number of miscarriages per 100 clinical pregnancies.

#### 2.2.10. Live Birth

The complete expulsion or extraction of a product of conception with evidence of life after 20 completed weeks of gestational age, or birth weight of 400 g or more if gestational age is unknown [30]. The live birth rate was calculated as the number of deliveries that resulted in at least one live birth per 100 embryo transfer cycles. The cumulative live birth rate (CLBR) represents the live births arising from ET and FET procedures, from a single oocyte pick-up (OPU).

### 2.3. Modalities of ART

#### 2.3.1. IVF-ICSI Split Modality (14.3% of Cycles)

Following our SDF studies with the Halosperm test reported firstly in 2016 [27] and a second study in 2021 [31], we have encouraged all new cases presenting to PIVET, especially those who have never had ART procedures elsewhere, i.e., IVF-naïve cases, to undergo an IVF-ICSI Split modality ensuring that all non-male factor cases have the best opportunity to achieve fertilisation and generate some embryos (avoiding the 5% chance of complete failed fertilisation and 15% chance of reduced fertilisation). A study which analysed four controlled and prospective trials, also found that those authors were unanimous in their conclusion that total fertilization failure of an IVF cycle can be prevented and fertilization can be improved if half of sibling oocytes are subjected to ICSI [32]. 

Following oocyte recovery, COCs were randomly divided into two groups—one group destined for IVF, the other destined for ICSI, the latter performed around 4–6 h post OPU following cumulus stripping (by hyaluronidase and mechanical pipetting) to reveal which of the oocytes have reached the MII stage. Ensuring the polar body is at the 12 o’clock position, the injection pipette is brought into line with zona pellucida at 3 o’clock position and the sperm is moved to the tip of the injection pipette. Thereafter the pipette is pushed through the zona pellucida into the centre of the mature oocyte where the single sperm is released. 

The group allocated to IVF have a sperm preparation of ~1 million/mL inseminated into the individual droplet of culture medium under paraffin oil which has been equilibrated overnight in the gassed environment of 5% CO_2_, 5% O_2_ and 90% N_2_, 4–6 h post OPU. The following morning at ~16–18 h post insemination all oocytes are denuded to reveal which are fertilized (identifying the 2PN stage with release of a second polar body). Other oocytes will be identified at the M-II stage (without pronuclei, indicating unfertilized mature oocyte) or immature stages (Metaphase-I or Geminal Vesicle oocytes). 

#### 2.3.2. ICSI-Only (ICSI) Modality (80.9% of Cycles)

Semen analysis profiles indicating sperm concentrations <5 million per mL and sperm morphology ratings <4% are the main indicators for allocating cases to ICSI. These parameters accord with the WHO Fifth Edition (2010) which remained the reference semen analysis for undertaking ART procedures during the course of this study. Male factor cases indicated for ICSI include those with oligozoospermia, asthenozoospermia or teratozoospermia, particularly those with the 3 defects (OAT syndrome); those with antispermatozoal antibodies in their semen (IgG levels > 20%) [33], those with elevation of SDF ≥ 15% [34], and those requiring surgical sperm retrieval including percutaneous epidydimal sperm aspiration (PESA), microsurgical epididymal sperm aspiration (MESA), and testicular sperm aspirations (both TESA and micro-TESE) as well as vasal flush procedures, mostly applied for males with spinal injury [35].

Clinical conditions may also be considered for ICSI, including males with varicoceles, previous orchidopexies or past genital trauma, particularly with reduced volume testes, chronic diseases, chronic drug utilizations and occupations with heavy metal exposure. Certain female factors are also encouraged to undertake ICSI, particularly where few oocytes are recovered (<4), or those with poorly explained infertility where intra-uterine insemination (IUI) treatments have failed and where either of the gametes (oocytes or spermatozoa) have been cryopreserved. Finally, those cases who have previously undertaken an IVF-ICSI Split modality for unexplained infertility will need little persuasion to undertake ICSI treatments, as over 25% of patients with unexplained infertility may have disordered zona pellucida-induced acrosomal reaction (DZPIAR) [36] and subsequently the ICSI fertilization rates were better than IVF-Only [31].

#### 2.3.3. IVF-Only (IVF) Modality (4.8% of Cycles)

Historically, the proportion of cases having the IVF-Only modality at PIVET has decreased markedly as the fertilization rates of oocytes within the IVF-ICSI Split modality has mostly been shown to be higher with the ICSI modality over an increasing range of indications as aforementioned. Furthermore, during this study period at PIVET, the overall fertilization numbers arising from the IVF-ICSI Split modality mean that the number of cases returning for repeat OPU cycles has reduced as the majority of cases have at least one blastocyst cryopreserved, causing a marked rise in FET cycles. During this study period, the proportion of cases having a single embryo transferred at either the fresh ET or the FET procedure has been of the order of 98%, again enhancing the number of cases having blastocysts cryopreserved, resulting in a marked reduction in the number of women requiring a further OPU procedure.

### 2.4. Sperm Preparation Techniques for ART

Spermatozoa need to be washed clean of seminal plasma before insemination in culture or in utero is performed. The washing usually involves some selection process whereby sperm with poor motility are minimized from the final preparation. This has the effect of improving the number of “normal” spermatozoa and yield a sperm population with minimal DNA damage. The choice of sperm preparation technique is determined by the characteristics of the sperm sample and its purpose. An ideal sperm preparation approach should recover a highly functional sperm population that preserves DNA and does not generate dysfunction through the production of reactive oxygen species either by sperm or from leukocytes [37]. Spermatozoa can survive for several days in culture media. Preparation of sperm was carried out in accordance with WHO 5th edition [37]. The culture medium used was ORIGIO^®^ sequential series (Cooper Surgical, Ballerup, Denmark) culture medium for IVF cases, whilst Quinn’s HEPES^®^ medium (SAGE, Trumbull, CT, USA) with added 5% human serum albumin (HSA) culture medium was used for ICSI cases.

#### 2.4.1. Direct Swim-Up

An assessment of sperm volume, count and motility was performed and recorded. An amount of 1 mL of semen was underlaid into an aliquot of 2 mL of culture medium and incubated in the incubator at 37 °C for 15 to 30 min. Following incubation, an aliquot of 1.5–2 mL of supernatant was removed from the test tube and the final pellet was re-suspended into a conical test tube which contained 1 mL of culture medium. Thereafter, it was centrifuged at 1500 rpm for 5 min. Most of the supernatant was removed and discarded, whilst the remaining pellet was re-suspended in 1 mL of medium. The final sperm preparation was diluted to 5 million/mL for IVF preparations or 1 million/mL for ICSI preparations. 

#### 2.4.2. Discontinuous Density Gradient—PureSperm^®^ 100

PureSperm^®^ (NidaCon International, Flöjelbergsgatan, Mölndal, Sweden) is the reagent used within the period of study. PureSperm^®^ gradients were warmed and maintained at 37 °C. An 80% solution of PureSperm^®^ was prepared by using 8 mL of PureSperm and 2 mL of Quinns HEPES buffer. A 40% solution of PureSperm^®^ was prepared by using 4 mL of PureSperm and 6 mL of Quinns HEPES buffer. Sperm volume, count and motility were assessed and recorded. The gradient consisted of 2 layers of 0.5 mL of PureSperm^®^: 80% and 40%. An aliquot of 1 mL of semen was pipetted onto the gradients and centrifuged at 1500 rpm for 15 min. The remaining pellet was removed carefully with a sterile glass Pasteur pipette and resuspended in 0.5–1 mL of culture medium in a conical test tube. Thereafter, it was centrifuged at 1500 rpm for 5 min. The pellet was washed once to remove the PureSperm^®^ and debris. The final sperm preparation was resuspended in 0.5–1 mL of culture medium. The final sperm preparation was diluted to 5 million/mL for IVF cases, and 0.1–1 million/mL for ICSI cases.

#### 2.4.3. Simple Sperm Washing

Severely oligozoospermic samples require a simplified sperm washing method in order to maintain sperm numbers. The sperm volume, count and motility were assessed and recorded. The semen sample was added into 1 mL of culture medium and centrifuged gently at 1500 rpm (gravitational force ~200× *g*) for 5 min. The supernatant was removed, and the remaining pellet was resuspended in 0.1–0.3 mL of culture medium. The final sperm preparation was diluted to 5 million/mL for IVF cases, and 0.1–1 million/mL for ICSI cases. Pentoxifylline was used if the motility was extremely poor.

#### 2.4.4. Frozen Semen Samples

One straw of frozen semen sample was removed from liquid nitrogen and thawed at 37 °C for ten minutes. Following thawing, the contents of the straw were placed into an empty tube. The sperm volume, count and motility were assessed and recorded. Based upon the post-thaw semen profile, the sperm was prepared by one of the above techniques. The final sperm preparation was diluted to 5 million/mL for IVF cases, and 0.1–1 million/mL for ICSI cases.

### 2.5. Sperm Chromatin Dispersion (SCD) Technique—Halosperm^®^

Halosperm^®^, patented by Halotech, was introduced at PIVET to measure sperm DNA fragmentation in 2013. The technique has been fully described [14,15] and validated against semen analysis profiles as well as male clinical parameters. Our validation studies indicated that the Halosperm test can only be reliably applied on semen samples with total sperm concentration of ≥5 million/mL. Our numerous studies indicate that adverse SDF levels are identified at ≥15% with advancing male age increasingly recording such levels. However, none of the other clinical parameters of stature, weight, or body mass index (BMI) showed any correlation. With respect to semen analysis profiles, high SDF levels are associated with prolonged abstinence period and raised semen volumes as well as poor motility patterns and abnormal sperm morphology, and are notably worse for tail defects [14]. 

The principle of Halosperm^®^ is based on a regulated DNA denaturation process that facilitates the subsequent removal of the proteins contained in each spermatozoon. In this approach, normal spermatozoa generate halos composed of DNA loops at the head of the sperm, whereas DNA-damaged spermatozoa produce small or no halos.

#### 2.5.1. Evaluation of DNA Damage

SDF was performed using the Halosperm^®^ G2 kit (Parque Cientifico de Madrid, Madrid, Spain). Halosperm^®^ cannot be performed on neat sperm concentrations of <5 × 10^6^ million/mL. Samples with counts between 5 and 19 million/mL should be concentrated by centrifuging at 2000 rpm for 10 min to obtain a suitable concentration for Halosperm testing. The neat sperm sample was diluted in isotonic sodium chloride buffer to a maximum concentration of 20 million sperm per mL. Then, 50 µL of each diluted semen aliquot was mixed with 100 µL of the melted agarose gel and placed in the 37 °C water bath to prevent gelification. An amount of 8µL of the sperm–agarose mixture was added to the labelled super-coated slides and covered with coverslips at room temperature (22 °C). The slides were then transferred to a refrigerator at 4 °C for 5 min. Post refrigeration, an initial acid treatment denatured the DNA in fragmented sperm cells, and thereafter a lysis solution treatment removed nuclear proteins and finally, washing, dehydration and multicoloured staining of the slides was performed at room temperature (22 °C).

The images of halos produced by Halosperm^®^ are strongly contrasted and can be precisely assessed under bright field microscope. A total of 200 spermatozoa cells were required to be assessed. It is feasible to distinguish spermatozoa from other cell types that may be present in the ejaculate, such as leukocytes, due to the preservation of their tails. Therefore, those cells which did not exhibit a clear tail were not included in the sperm count for DNA fragmentation. In addition, Halosperm^®^ enables the visualisation of spermatozoa with significantly damaged DNA relative to other types of less severe forms of damage. Those spermatozoa observed with a halo width similar to or <1/3 of the diameter of the core, weakly stained or without halo were classified as spermatozoa with DNA fragmentation. Results were expressed as a percentage of spermatozoa cells with DNA fragmentation. The diagnostic sensitivity and specificity are 93%. Figure 2 depicts the varying degree of halo dispersion and sperm without halo.

#### 2.5.2. Validation and Quality Control

The Halosperm^®^ test results of DFI were compared with controls and replicated by separate embryologists. If the values exceeded a permissible range of 25%, the test would be repeated. Our experience revealed that the negative controls (without or minimal halo) showed differences of ±8.0%, due to the subjective interpretation of small or no halo, while positive controls (with halo) showed differences ±1.6%. The embryologists who performed the Halosperm test receive quality control assessment from QAP Online (FertAid Pty Ltd.; Newcastle, NSW, Australia) which operates an international Internet site dedicated to the provision of quality assurance and training in the Reproductive Sciences, and it monitors many aspects of the embryology and andrology within PIVET laboratory.

### 2.6. Statistical Analysis

The statistical analyses of demographic data were performed using SPSS software (version 26.0, SPSS Inc.). All numeric data are presented as the mean value ± standard deviation, analysed by Student’s t–test. The laboratory and clinical outcomes were compared between the two DFI groups being analysed by Chi-squared or Fisher’s exact test. The rates of ET and FET were also compared between the two DFI groups. Multivariate logistic regression was performed to determine the confounders. Differences between the values were considered statistically significant when 2-sided *p* < 0.05. 

## 3. Results

The baseline characteristics including the causes of infertility are shown in Table 1. The mean of SDF in the <15% and ≥15% groups were 7.43 and 25.24, respectively. The advancing male age was observed in the SDF ≥ 15% group (36.62 vs. 37.64, *p* < 0.01). The duration of infertility of the couple was found to be significantly longer (*p* < 0.0001), and significantly higher dosages of gonadotrophins were needed for stimulation in the group of SDF ≥ 15% (*p* < 0.0001). The majority of the males provided a fresh ejaculate (92%) and 8% of the males relied on a frozen sample. The discontinuous density gradient was the preferred technique in SDF ≥ 15% (65.4% vs. 62.6%, *p* < 0.0001) due to the nature of the semen sample which comprises more abnormality parameters (less progressive motility, morphology and concentration) compared to SDF < 15%. A comparison of seminal variables in both groups is presented in Table 2. 

### 3.1. Impact of SDF Levels on Laboratory and Clinical Outcomes with Female Age Groups Stratification

This cohort included 1148 couples who underwent a total of 1600 IVF/ICSI cycles, retrieving 12,056 oocytes, and leading to 2813 transferred embryos in 2554 cycles (1396 were fresh and 1158 were cryopreserved embryos). From these transfers, 640 babies were delivered (287 were from fresh and 353 were from cryopreserved embryos). The laboratory and clinical outcomes based on SDF levels among female age groups are shown in Table 3. In general, there were no significant differences observed in terms of the fertilization, blastocyst and good-quality blastocyst rates in both SDF groups. Nevertheless, our study found that while the fertilization rate per oocyte treated was similar in both SDF groups, the fertilization rate per MII oocyte (adjusted fertilization rate) was significantly higher in the group of SDF < 15% (76% vs. 73.5%, *p* = 0.02). The group of SDF ≥ 15% were found to have significant lower good-quality blastocyst rates with women ≥45 years (6.4% vs. 22.9%, *p* = 0.74) when compared to the group of SDF ≥ 15%. 

With respect to clinical pregnancy, miscarriage and live birth rates, no differences were observed in either the SDF <15% or ≥15% group. Nevertheless, when stratifying the women’s age group, men with SDF < 15% generated a significantly higher clinical pregnancy and live birth rate with women <35 years compared to their counterparts (44.1% vs. 37.4%, *p* = 0.04; 38.6% vs. 28.6%, *p* < 0.01). The miscarriage rate was higher and live birth rate was relatively lower in the SDF ≥ 15% group, albeit these differences were not significant. 

### 3.2. Impact of SDF Levels on Laboratory and Clinical Outcomes among Three ART Modalities

Among the three modalities, ICSI displayed the best fertilization rate in both SDF <15% and ≥15% groups (75.5% and 74.1%, *p* < 0.0001 for both). However, with respect to the adjusted fertilization rate, ICSI remained significantly higher for SDF ≥ 15% (*p* = 0.03), but not significant in the group of SDF < 15% (*p* = 0.08). The best blastocyst rate was displayed with IVF in the SDF < 15% group (57.3%, *p* < 0.0001), whilst IVF-ICSI Split generated the highest blastocyst rate in the SDF ≥ 15% group (63.1%, *p* < 0.001). IVF-ICSI Split showed the highest rate of generating good-quality blastocysts in both <15% and ≥15% SDF groups (44.9%, *p* < 0.0001; 50.0%, *p* < 0.001). It is relevant to mention that the IVF normal fertilization rate (71.8%), ICSI normal fertilization rate (75.7%), blastocyst development rate (46.4%) and good-quality blastocyst rate (38.4%) during this study period were above the competency value in accordance with the Vienna consensus by ESHRE [29]. The highest proportion of clinical pregnancies and live birth with a lower miscarriage rate, arose from the IVF-ICSI Split group with SDF < 15%. The highest miscarriage rate occurred in ICSI fresh-transferred pregnancies in the SDF < 15% group (*p* < 0.01). There were no differences observed in respect to clinical, miscarriage and live birth rates among the three modalities in the SDF ≥ 15% group. The summary of laboratory and clinical outcomes based on SDF levels among the three ART modalities are depicted in Table 4. 

### 3.3. Impact of SDF Levels on Live Birth Rates Stratified by Women’s Age Group and by Three ART Modalities

In addition, we also performed an analysis of live birth rates, being the ultimate goal in ART, in both SDF groups according to the women’s age groups and among the three ART modalities. From this study, we found that IVF cycles have the best overall live birth for all age groups in the SDF < 15% group, especially in women below 40 years. There were no differences observed among the three modalities in the SDF ≥ 15% group, except IVF displayed the best live birth rate in women aged 35–39 years (50%, *p* = 0.04). The details are presented in Table 5 and Figure 3.

### 3.4. Subgroup Analysis of the Impact of SDF Levels on Laboratory and Clinical Outcomes within IVF-ICSI Split Cycles

Given the outcomes for the highest quality blastocyst and good-quality blastocyst in both SDF groups, the highest clinical pregnancy and live birth rates occurred in the IVF-ICSI Split group in SDF < 15%. We therefore further undertook a subgroup analysis within the IVF-ICSI Split cycles. The best fertilization rate was observed in the ICSI arm in both the SDF groups (80.3% and 76.4%, both *p* < 0.0001). However, the IVF arm displayed the highest blastocyst rate in the SDF 15% group (59.3%, *p* = 0.03) to a significant degree. In terms of good-quality blastocyst rates, there were no differences observed in both groups. In the SDF < 15% group, the clinical pregnancy rate was optimal in the IVF group (47.6%, *p* = 0.03). No differences were observed for the live birth rate between the SDF groups. None of the modalities was superior to each other in the SDF ≥ 15% group in terms of the clinical outcomes. The summary of the results is presented in Table 6.

### 3.5. Multivariate Logistic Regression Analysis of Clinical Variables Associated with Clinical Outcomes

Due to the lack of an overall difference in clinical outcomes between the two SDF groups, it appears that SDF levels may not be the sole determinant of ART success. Therefore, we subsequently evaluated the independently significant variables correlated to clinical outcomes via multivariate logistic regression analysis. This was in order to understand the complex interplay between SDF levels and other factors that may influence the ART outcomes. 

The summary of results is shown in Table 7. Younger female age was found to be a strong predictor in determining the success of clinical pregnancy (Odds Ratio; OR 0.90, 95% CI 0.83–0.98) and cumulative live birth rate (OR 0.86, 95% CI 0.80–0.94). No significant difference was observed between the ovarian reserve measured by AMH and the clinical outcomes. The total mature oocytes retrieved and sperm preparation method (favouring discontinuous density gradient, OR 0.86, 95% CI 0.74–0.99) significantly influence the outcome of the miscarriage rate. Increased BMI of both female and male (OR 1.16, 95% CI 1.06–1.27; OR 1.01, 95% CI 1.00–1.03) was also shown to be associated with increased miscarriage rate. Surprisingly, advancing age in the male is associated with lower miscarriage rate (OR 0.90, 95% CI 0.82–0.98). 

## 4. Discussion

The impact of sperm DNA damage on ART pregnancy has been the subject of numerous studies. In this study, we found that the rates of unadjusted fertilization, blastocyst development and good-quality blastocyst rates were comparable in both SDF groups. This is consistent with a study in Greece in which the SDF was also assessed by Halosperm; the authors did not observe any correlation between the SDF and fertilization, cleavage rate and embryo quality at the SDF cut off of 35% [38]. However, applying the adjusted fertilization rate, we recognized a significant difference according to the SDF levels such that the group of SDF < 15% has significantly higher fertilization rates whilst the group of SDF ≥ 15% registers a significantly poorer fertilization rate, especially marked for younger women (<35 years). Several other clinical studies have also not found any significant difference between sperm DNA fragmentation and fertilization rates in vitro [17,39] but the methodologies in these studies failed to describe any adjustment for insemination vs ICSI where COCs have MII maturity identified before fertilization. Furthermore, failure to detect different fertilization rates (either unadjusted or adjusted) according to SDF levels may not be completely unexpected as the clinical implications of sperm DNA damage for the embryo may be delayed, due to the peri-fertilization stage being under parental influence. There may be a stronger maternal influence on fertilization while the paternal genome exerts more influence on the embryo development during later stages [40]. 

Notwithstanding the above, some studies display contrary findings to ours. One study from Guangxi reported on women aged ≤35 years who generated 1152 embryos by ICSI where SDF was assessed by SCD, and found that the fertilization rate and blastocyst quality were significantly higher in the SDF < 15% group than in the SDF ≥ 15% group [41]. Whilst our study agreed that adjusted fertilization rates were significantly higher in the group with SDF < 15%, we remain unable to explain why our study did not show similarly higher blastocyst rates with SDF < 15%. The absence of smoking history and lifestyle among the males may be a limitation for the interpretation of these dissimilar findings, requiring future consideration. 

In another study, being a prospective cohort analysis of 475 ICSI cycles in which sperm DNA damage was evaluated by the SCD test, the authors observed that elevated sperm DNA fragmentation was associated with a slower cleavage rate, poor Day 3 embryo quality, and poor blastocyst development and quality, but not with a lower fertilization rate [42]. These contrary findings are probably due to the ‘late paternal effect’ on embryo development [43] as shown in numerous studies demonstrating that high SDF can slow down the morphokinetic parameters and developmental potential of embryos [41,42,44]. The earliest stages of embryo development in mammalian species, including humans, are regulated at a post-transcriptional level by maternally inherited information; consequently, the impact of sperm DNA damage on embryo development is expected to become apparent at the four-cell to eight-cell stage, when expression of the zygotic genome begins [45,46]. Nevertheless, our study has failed to show this late paternal effect when comparing the blastocyst development rate in both SDF groups. 

In this present study, we found that males with SDF < 15% generally present a favourable trend in respect to clinical and live birth rate outcomes, and with a lower miscarriage rate compared with males with SDF ≥ 15%, albeit statistically they did not reach significance. Nevertheless, when comparing these outcomes among the four age groups of the women, we found that the clinical and live birth rates were significantly higher among females aged <35 years where their male partners had SDF < 15%. Female age alone is inversely related to fertility outcome, mostly as a result of the large increase in aneuploidy and spontaneous miscarriage rates with increasing maternal age [47]. Furthermore, spermatozoa are incapable of DNA repair, and they rely on the oocyte for repair after fertilisation. The degree to which sperm DNA damage affects pregnancy outcome may depend on oocyte quality to some extent, and the significance of oocyte age for the capacity to repair sperm DNA damage has been confirmed using an animal model [48]. In addition, the negative impact of excessive DNA fragmentation on pregnancy can be overcome by using good-quality oocytes, as demonstrated by a study comparing standard cycles to donor cycles, where the authors found younger oocytes preserved DNA repair ability that can withstand up to 40% sperm chromatin damage and still result in a normal pregnancy [49]. 

When analysing our secondary outcomes of comparing the three ART modalities, there were no significant differences in the adjusted fertilization rates within the <15% SDF group, but ICSI displayed a significantly higher fertilization rate within the ≥15% SDF group. This finding was also shown for the ICSI component of the IVF-ICSI Split modality within the ≥15% SDF group. This is in line with a few other studies which found that the proportion of fertilization is greater with ICSI than with IVF [15,50,51,52], albeit that the fertilization rates were not always adjusted for the IVF modality. A meta-analysis evaluating the relationship between sperm DNA damage and fertilization rate using either SCSA or TUNEL assays actually reported no significant differences during IVF or ICSI [53]. However, the occurrence of total fertilisation failure and poor fertilisation were found to be higher with IVF [51]. On the contrary, another study involving 745 women aged 40–43 years, who underwent IVF or ICSI treatments for non-male factor infertility showed that the fertilization rate and fertilization failure were similar in IVF or ICSI treatment [54]. Surprisingly, a more recent retrospective cohort study from China involving 549 IVF and 242 ICSI cycles for individuals with unexplained infertility found that although ICSI was not associated with a high fertilization rate, it did have a higher cancellation rate of fresh embryo transfer, owing to a larger proportion of them having poor quality embryos [55]. 

In our study, the higher fertilization rate in the ICSI group, however, did not translate into an improved good-quality blastocyst rate, neither for better clinical nor live birth rates. In the group of SDF < 15%, ICSI has the lowest blastocyst and good-quality blastocyst rates, and lowest clinical and live birth rates, with the highest miscarriage rate from fresh ETs. These data were concordant with several other studies, in which the authors found that ICSI offers no advantage over IVF in terms of fertilization, implantation and pregnancy rates in non-male-factor infertility and therefore they supported the notion that ICSI should be reserved for severe male-factor infertility [56,57,58]. A large population-based retrospective cohort involving 14,693 women in Victoria, Australia, demonstrated that ICSI resulted in a lower fertilization rate per oocyte retrieved and a similar cumulative live birth rate, compared to IVF [59]. Similarly, another large cohort study in the UK using the wealth of the Human Fertilization and Embryology Authority (HFEA) database, strongly suggests no benefit in the clinical pregnancy rate or live birth outcome with ICSI in all autologous ovarian response cycles with a normospermic profile [52]. In the SDF group of <15%, even though the IVF modality did not display the best clinical pregnancy rate, it has the lowest miscarriage rate and highest live birth rate among women aged <40 years. This finding is once again consistent with the aforementioned studies, and in keeping with the recommendation of the practice committee of ASRM, who found no data to support the routine use of ICSI for non-male factor infertility [60]. They recommended ICSI may be beneficially extended for patients using pre-implantation genetic testing, in-vitro maturation or cryopreserved oocytes [60].

There exists controversial data concerning the ART modalities and clinical outcomes with high SDF in several meta-analyses, owing to the included studies having variable study characteristics, different sperm fragmentation assays, different thresholds for DNA fragmentation and heterogeneity, including the presence of both retrospective and prospective studies. Two meta-analyses concluded that no association between high DNA fragmentation and miscarriage with different ART treatments could be found [15,21]. Another meta-analysis indicated that high SDF was related to higher miscarriage rates in both IVF and ICSI cycles, whereas it was associated with lower pregnancy rates in IVF but not in ICSI cycles [22]. Furthermore, a study involving 1633 participants in Sweden showed that SCSA levels with SDF ≥ 20% had a lower live birth rate with IVF but not with ICSI treatment. One study discovered that even for ICSI, there was a tendency for higher pregnancy rates with SDF > 30% rather than with SDF < 30%, and ICSI was therefore the more effective treatment approach than IVF [61]. Furthermore, they also found that sperm DNA damage was not associated with an increased risk of early pregnancy loss in women <40 years at a DFI threshold value of ≥30% [61]. Nonetheless, in our present study, we found that none of the ART modalities was superior to each other in the group of SDF ≥ 15% in terms of clinical pregnancy rates, miscarriage rates or live birth rates. This is consistent with a study which reported that the implantation, miscarriage and pregnancy rates were similar in IVF and ICSI [55], albeit it was noteworthy that there was a mismatch in the number of cases among the three ART modalities in the group of ≥15%, which may contribute to these findings.

In this study, female age was found to be a strong determinant in optimizing the clinical and cumulative live birth rate, which is in line with many studies mentioning infertility increases with age and the affect on the success rates of infertility treatments [62,63,64]. We also found that AMH has no influence on these outcomes, reflecting the importance of quality of oocytes rather than the quantity. Given the fact that female age is non-modifiable, clinicians should focus on the SDF level, which is reversible. The sperm DNA damage can be improved by several strategies, such as lifestyle modification, reduced abstinence, weight optimization, institution of antibiotics for male genital infections, repair of clinical varicocele, sperm selection techniques and usage of testicular sperm in obstructive azoospermia [65]. In addition, current guidelines and expert opinions from a global survey on the management of elevated SDF in male infertility recommended the prescription of empirical antioxidants for infertile men with elevated SDF, in particular with those associated with idiopathic infertility, recurrent pregnancy loss, varicocele, leukocytospermia, smoking or other lifestyle and environmental risk factors [66]. Although a duration of 3–6 months has been proven successful, there is no unanimous recommendation on the type, dosage, and duration of antioxidant treatment. The success of treatment should be guided by improved conventional semen analysis, decreased SDF levels, and improved reproductive outcomes, by either natural or ART conception [66].

The main limitation of this presented study is that it is retrospective in nature, where the data were reliant on record keeping and data entry. Furthermore, the clinical setting is entirely private, with patients paying substantial fees for ART services, therefore with a view to obtaining the best possible outcomes for the patient, in the group of SDF ≥ 15%, most of the patients were encouraged to utilise ICSI despite the higher cost. Other cases undertook ICSI based on a higher fertilization rate from a previous IVF-ICSI Split cycle; therefore there were reduced numbers of cases for analysis in the IVF and subsequent IVF-ICSI Split groups. These factors therefore led to mismatched numbers among the groups, with the ratio of 4:1:16 for IVF-ICSI Split, IVF and ICSI groups, affecting the statistical evaluation for interpretation of the data. The strength of this study is that we not only compared reproductive outcomes at high and low SDF levels, but also stratified them according to women’s age groups and ART modalities, at which the woman’s age has shown to be the significant independent variable associated with clinical outcomes and its relevant to clinical practice. 

## 5. Conclusions

Our primary outcome showed the SDF assay is predictive of clinical and live birth rates in women <35 years, where the clinical and live birth rate were significantly higher in the group of SDF < 15% when compared to SDF ≥ 15% group. For the secondary outcome, although the clinical pregnancies were not the highest in IVF, it has the lowest miscarriage rate and highest live birth rate in SDF < 15% with women aged below 40 years. Therefore, IVF-Only is a promising modality in the SDF < 15% group in women below 40 years, when compared to the SDF ≥ 15% group. In the group of SDF ≥ 15%, the IVF-ICSI Split cycle exhibited the highest blastocyst and high-quality blastocyst rates, with no significant differences between IVF and ICSI treatment within the split group. In conclusion, the SDF assay was found to be predictive of the adjusted fertilization rate for women <35 years, significantly predictive of clinical pregnancy as well as the live birth rate for women aged <40 years utilising the IVF modality and whose male partner has SDF < 15%, bearing in mind that the IVF case numbers were disproportionately low. Future well designed prospective research is required to confirm the effect of SDF on different ART modalities and with the ART outcomes. Notwithstanding the current view that science cannot beat the biological clock, research on the idea of ovarian rejuvenations should also be considered in the distant future with a view to improving oocyte qualities.

## Figures and Tables

**Figure 1 jpm-13-01079-f001:**
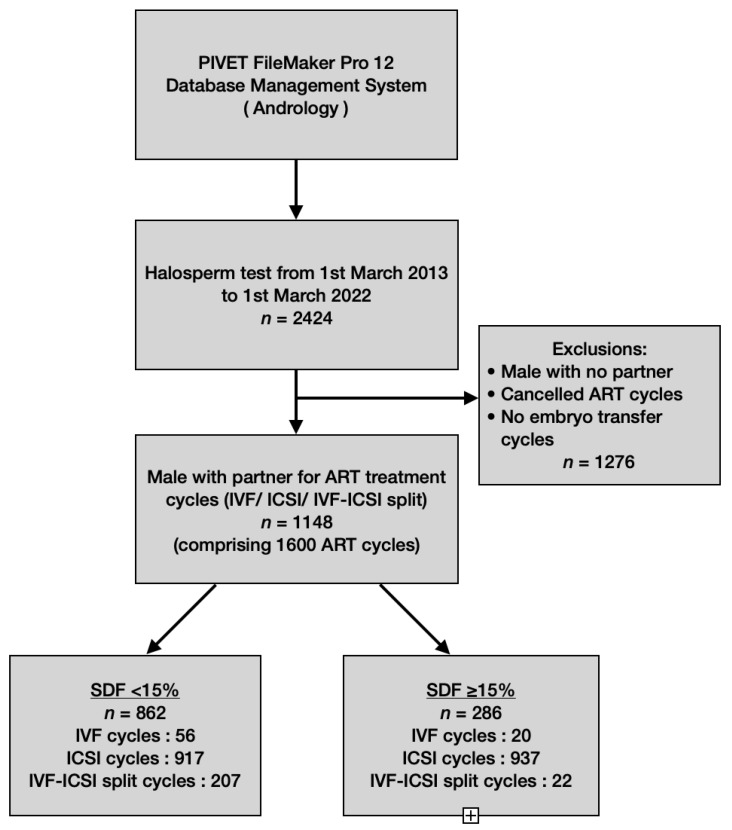
Flow chart of data extraction.

**Figure 2 jpm-13-01079-f002:**
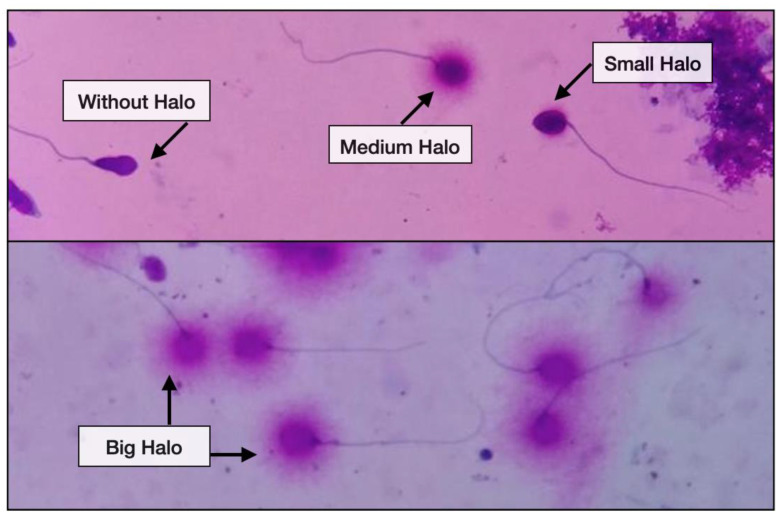
The varying degree of halo dispersion and without halo.

**Figure 3 jpm-13-01079-f003:**
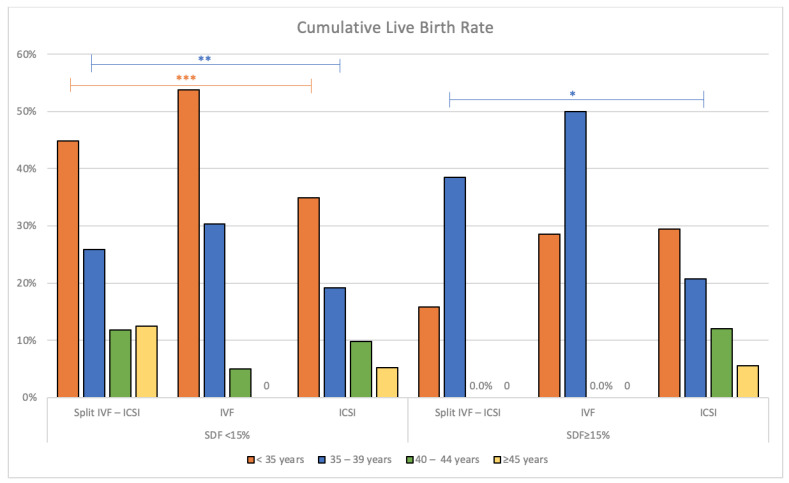
Cumulative live birth rate (fresh and frozen transfer) per embryo transfer cycle based on SDF levels <15% and ≥15%, and the woman’s age group among three modalities. 0: nil of case. *** *p*-value < 0.01; ** *p*-value = 0.03; * *p*-value = 0.04.

**Table 1 jpm-13-01079-t001:** Baseline characteristics undertaken during workup for the initiated ART cycle presented with respect to low vs. high SDF levels.

Variables Initiated Cycle	SDF < 15% n = 1180	SDF ≥ 15% n = 420	*p* Value
Male age (years)	36.62 ± 6.72	37.64 ± 6.25	<0.01 ^a^
Male BMI (kg/m^2^)	27.32 ± 4.43	26.95 ± 4.42	0.20 ^a^
Female age (years)	35.94 ± 5.37	35.94 ± 4.89	0.70 ^a^
Female BMI (kg/m^2^)	24.47 ± 4.67	24.88 ± 4.67	0.12 ^a^
AMH (pm/L)	19.04 ± 19.52	17.93 ± 19.04	0.35 ^a^
Infertility duration (years)	2.65 ± 2.12	3.09 ± 2.88	<0.0001
Gonadotrophin total administered (IU)	3030.51 ± 2038.08	3193.73 ± 2308.46	<0.0001 ^a^
Estradiol level at trigger (pmol/L)	7886.54 ± 4833.72	7728 ± 4338.73	0.56 ^a^
Day of trigger	13.04 ± 4.21	13.42 ± 4.07	0.06 ^a^
Day of OPU	15.04 ± 4.21	15.42 ± 4.07	0.06 ^a^
Oocytes retrieved per cycle (n)	9.50 ± 5.93	9.40 ± 6.13	0.18 ^a^
Mature oocytes per cycle (n)	7.22 ± 4.75	6.98 ± 4.86	0.69 ^a^
Type of ejaculate			0.78 ^b^
Fresh ejaculate	91.3%	92.8%	
Frozen sperm	8.7%	6.2%	
Sperm preparation technique			
Discontinuous density gradient	62.6%	65.4%	<0.0001 ^b^
Direct swim up	34.9%	26.1%	<0.001 ^b^
Simple Sperm Washing	2.4%	8.3%	0.30 ^b^
Density gradient + swim up	0.1%	0.2%	0.50 ^b^
Causes of infertility			–
Endometriosis	4.3%	2.9%	
Tubal factor	8.5%	3.9%	
Diminished ovarian reserve	7.4%	3.9%	
Male factor	20.9%	33.8%	
Unexplained infertility	38.2%	20.0%	
Male and female factors	10.6%	29.4%	
Vasectomy/reversal	0.4%	–	
PCOS	5.0%	2.6%	
Cancer/chemotherapy	0.1%	–	
POI	0.3%	–	
Fibroid/Adenomyosis	1.6%	–	
Anovulatory	0.4%	0.4%	
Others	2.2%	3.1%	
Antral Follicle Count			–
A++ (≥40 follicles)	6.3%	9.6%	
A+ (30–39 follicles)	5.6%	7.2%	
A (20–29 follicles)	13.5%	6.8%	
B (13–19 follicles)	22.2%	20.0%	
C (9–12 follicles)	21.3%	18.4%	
D (5–8 follicles)	17.9%	21.3%	
E (≤4 follicles)	10.7%	12.1%	
Not recorded	2.5%	4.6%	

Numerical data presented in mean ± standard deviation. *p* value was calculated by ^a^ Student’s *t* test or ^b^ Chi–squared test.

**Table 2 jpm-13-01079-t002:** Male seminal characteristics categorized according to DNA fragmentation with SDF <15% and ≥15% groups.

Seminal Variables	SDF < 15%	SDF ≥ 15%	*p*-Value
SDF (%)	7.43 ± 3.41	25.24 ± 11.11	<0.0001
Volume (mL)	3.50 ± 1.55	3.70 ± 1.58	0.02
pH	8.06 ± 0.27	8.03 ± 0.26	0.16
Abstinence (days)	4.25 ± 5.10	4.83 ± 4.52	0.04
Concentration (10^6^/mL)	66.42 ± 52.82	56.99 ± 53.11	<0.01
Normal morphology (%)	5.61 ± 3.32	4.91 ± 3.49	<0.001
Total motility (%)	63.80 ± 16.17	56.18 ± 19.56	<0.0001
Progressive motility (%)	58.07 ± 16.00	50.31 ± 19.18	<0.0001

**Table 3 jpm-13-01079-t003:** Laboratory and clinical outcomes per cycle reaching OPU, categorized according to SDF groups <15% and ≥15% and female age groups.

	SDF < 15%	SDF ≥ 15%	*p*-Value
Age (years)	Laboratory Outcome		
Fertilization rate <35 35–39 40–44 ≥45	6457/8979 (71.9%) 3165/4349 (72.8%) 2078/2852 (72.9%) 1070/1570 (68.2%) 144/208 (69.2%)	2202/3077 (71.6%) 1050/1488 (70.6%) 743/1020 (72.8%) 362/500 (72.4%) 47/69 (68.1%)	0.71 ^a^ 0.10 ^a^ 0.99 ^a^ 0.07 ^a^ 0.88 ^b^
Adjusted Fertilization rate <35 35–39 40–44 ≥45	6457/8532 (76.0%) 3165/4072 (78.4%) 2078/2730 (76.1%) 1070/1526 (70.1%) 144/204 (70.6%)	2202/2996 (73.5%) 1050/1439 (73.0%) 743/995 (74.7%) 362/493 (73.4%) 47/69 (68.1%)	0.02 ^a^ <0.0001 ^a^ 0.36 ^a^ 0.16 ^a^ 0.76 ^b^
Blastocyst rate <35 35–39 40–44 ≥45	2965/6457 (45.9%) 1699/3165 (53.7%) 918/2078 (44.2%) 315/1070 (29.4%) 33/144 (22.9%)	1053/2202 (47.8%) 562/1050 (53.5%) 370/743 (49.8%) 116/362 (32.0%) 5/47 (10.6%)	0.12 ^a^ 0.93 ^a^ <0.01 ^a^ 0.35 ^a^ 0.09 ^b^
Good-quality blastocyst rate <35 35–39 40–44 ≥45	2476/6457 (38.3%) 1399/3165 (44.2%) 764/2078 (36.8%) 281/1070(26.3%) 32/144 (22.2%)	853/2202 (38.7%) 455/1050 (43.3%) 299/743(40.2%) 96/362(26.5%) 3/47 (6.4%)	0.74 ^a^ 0.62 ^a^ 0.09 ^a^ 0.92 ^a^ 0.02 ^b^
	Clinical Outcome		
Clinical pregnancy rate <35 35–39 40–44 ≥45	590/1888 (31.3%) 359/814 (44.1%) 163/610 (26.7%) 60/398 (15.0%) 8/66 (12.1%)	204/666 (30.4%) 110/294 (37.4%) 75/243 (30.9%) 17/111 (15.3%) 2/18 (11.1%)	0.77 ^a^ 0.04 ^a^ 0.22 ^a^ 0.95 ^a^ 1.0 ^b^
Miscarriage rate <35 35–39 40–44 ≥45	173/590 (29.3%) 82/359 (22.8%) 57/163 (35.0%) 30/60 (50.0%) 4/8 (50.0%)	64/204 (31.4%) 35/110 (31.8%) 23/75 (30.7%) 5/17 (29.4%) 1/2 (50.0%)	0.58 ^a^ 0.06 ^a^ 0.66 ^b^ <0.001 ^b^ 1.0 ^b^
Cumulative livebirth rate <35 35–39 40–44 ≥45	487/1888 (25.8%) 314/814 (38.6%) 130/610 (21.3%) 39/398 (9.8%) 4/66 (6.1%)	153/666 (22.5%) 84/294 (28.6%) 55/243 (22.6%) 13/111 (11.7%) 1/18 (5.6%)	0.15 ^a^ <0.01 ^a^ 0.67 ^a^ 0.56 ^a^ 1.0 ^b^

*p*-value was calculated by ^a^ Chi-squared test or ^b^ Fisher’s exact test.

**Table 4 jpm-13-01079-t004:** Laboratory and clinical outcomes per cycle based on SDF levels <15% and ≥15% among three ART modalities.

	SDF < 15%			*p*-Value	SDF ≥ 15%			*p*-Value	
ART Modality Initiated Cycles	Split n = 207	IVF n = 56	ICSI n = 917		Split n = 22	IVF n = 20	ICSI n = 378		
Laboratory Outcome									
Fertilization rate	1651/2488 (66.4%)	379/626 (60.5%)	4427/5865 (75.5%)	<0.0001 ^a^	160/267 (59.9%)	134/235 (57.0%)	1908/2575 (74.1%)	<0.0001 ^a^	
Adjusted Fertilization rate	1651/2187 (75.5%)	379/480 (79.0%)	4427/5865 (75.5%)	0.08 ^a^	160/237 (67.5%)	134/184 (72.8%)	1908/2575 (74.1%)	0.03 ^a^	
Blastocyst rate	926/1651 (56.1%)	217/379 (57.3%)	1822/4427 (41.2%)	<0.0001 ^a^	101/160 (63.1%)	63/134 (47.0%)	889/1908 (46.6%)	<0.001 ^a^	
Good-quality blastocyst rate	742/1651 (44.9%)	161/379 (42.5%)	1573/4427 (35.5%)	<0.0001 ^a^	80/160 (50.0%)	36/134 (26.9%)	737/1908 (38.6%)	<0.001^a^	
Clinical Outcome									
Clinical pregnancy rate									
ET	70/180 (38.9%)	10/47 (21.3%)	187/827 (22.6%)	<0.0001 ^a^	4/17 (23.5%)	6/15 (40%)	70/310 (22.6%)	0.12 ^a^	
FET	106/231 (45.9%)	17/45 (37.8%)	200/558 (35.8%)	<0.01 ^a^	9/17 (52.9%)	7/16 (43.8%)	108/291 (37.1%)	0.46 ^a^	
Miscarriage rate									
ET	15/70 (21.4%)	0/10 (0%)	72/187 (38.5%)	<0.01 ^b^	3/4 (75%)	1/6 (16.7%)	26/70 (37.1%)	0.18 ^b^	
FET	28/106 (26.4%)	6/17 (35.3%)	52/200 (26.0%)	0.40 ^a^	3/9 (33.3%)	2/7 (28.6%)	29/108 (26.9%)	0.90 ^b^	
Cumulative livebirth rate									
ET	63/180 (35.0%)	14/47 (29.8%)	147/827 (17.8%)	<0.0001 ^a^	2/17 (11.8%)	5/15 (33.3%)	56/310 (18.1%)	0.10 ^a^	
FET	81/231 (35.1%)	18/45 (40.0%)	164/558 (29.4%)	0.04 ^a^	6/17 (35.3%)	5/16 (31.3%)	79/291 (27.1%)	0.43 ^a^	

*p*-value was calculated by ^a^ Chi-squared test or ^b^ Fisher’s exact test.

**Table 5 jpm-13-01079-t005:** Clinical outcome per embryo transfer according to DFI levels <15% and ≥15% within age groups and among three ART modalities.

	SDF < 15%			*p*-Value	SDF ≥ 15%			*p*-Value
ART Modality	Split	IVF	ICSI		Split	IVF	ICSI	
Age (Years)	Clinical Pregnancy Rate							
<35	118/230 (51.3%)	17/39 (43.6%)	224/545 (41.1%)	0.03 ^a^	7/19 (36.8%)	9/21 (42.9%)	94/254 (37.0%)	0.03 ^b^
35–39	51/139 (36.7%)	9/33 (27.3%)	103/438 (23.5%)	<0.01 ^a^	6/13 (46.1%)	4/8 (50.0%)	65/222 (29.3%)	0.22 ^b^
40–44	5/34 (14.7%)	1/20 (5%)	54/344 (15.7%)	0.43 ^a^	0/2 (0%)	0/2 (0%)	17/107 (15.9%)	1.0 ^b^
≥45	2/8 (22.0%)	–	6/58 (10.3%)	0.25 ^b^	–	–	2/18 (11.1%)	–
Miscarriage rate								
<35	26/118 (22.0%)	4/17 (23.5%)	52/224 (23.2%)	0.97 ^a^	5/7 (71.4%)	3/9 (33.3%)	27/94 (28.7%)	0.09 ^b^
35–39	15/51 (29.4%)	2/9 (22.2%)	40/103 (38.8%)	0.45 ^a^	1/6 (16.7%)	0/4 (0%)	22/65 (33.8%)	0.40 ^b^
40–44	1/5 (20.0%)	0/1 (0%)	29/54 (53.7%)	0.19 ^b^	0/0 (0%)	0/0 (0%)	5/17 (29.4%)	1.0 ^b^
≥45	1/2 (50.0%)	–	3/6 (50.0%)	1.0 ^b^	–	–	1/2 (50.0%)	–
Live birth rate								
<35	103/230 (44.8%)	21/39 (53.8%)	190/545 (34.9%)	<0.01 ^a^	3/19 (15.8%)	6/21 (28.6%)	75/254 (29.5%)	0.48 ^b^
35–39	36/139 (25.9%)	10/33 (30.3%)	84/438 (19.2%)	0.03 ^a^	5/13 (38.5%)	4/8 (50.0%)	46/222 (20.7%)	0.04 ^b^
40–44	4/34 (11.8%)	1/20 (5%)	34/344 (9.8%)	0.41 ^a^	0/2 (0%)	0/2 (0%)	13/107 (12.1%)	1.0 ^b^
≥45	1/8 (12.5%)	–	3/58 (5.2%)	0.41 ^b^	–	–	1/18 (5.6%)	–

*p*-value was calculated by ^a^ Chi-squared test or ^b^ Fisher’s exact test.

**Table 6 jpm-13-01079-t006:** Laboratory and clinical outcomes per cycle according to SDF levels <15% and ≥15% for IVF-ICSI Split modality.

ART Modality	SDF < 15%		*p*-Value	SDF ≥ 15%		*p*-Value
	IVF	ICSI		IVF	ICSI	
Laboratory Outcome						
Fertilization rate	702/1306 (53.8%)	949/1182 (80.3%)	<0.0001 ^a^	66/144 (45.8%)	94/123 (76.4%)	<0.0001 ^b^
Adjusted Fertilization rate	702/1005 (69.9%)	949/1182 (80.3%)	<0.0001 ^a^	66/114 (57.9%)	94/123 (76.4%)	<0.01 ^b^
Blastocyst rate	416/702 (59.3%)	510/949 (53.7%)	0.03 ^a^	41/66 (62.1%)	60/94 (63.8%)	0.87 ^b^
Good– quality blastocyst rate	326/702 (46.4%)	416/949 (43.8%)	0.29 ^a^	33/66 (50.0%)	47/94 (50.0%)	1.0 ^b^
Clinical Outcome						
Clinical pregnancy rate						
ET	40/84 (47.6%)	30/96 (31.3%)	0.03 ^b^	3/6 (50.0%)	1/11 (9.1%)	0.10 ^b^
FET	40/88 (45.5%)	66/143 (46.2%)	1.0 ^b^	3/6 (50.0%)	6/11 (54.5%)	0.99 ^b^
Miscarriage rate						
ET	8/40 (20.0%)	7/30 (23.3%)	1.0 ^b^	1/3 (33.3%)	1/1 (100%)	1.0 ^b^
FET	11/40 (27.5%)	17/66 (25.8%)	1.0 ^b^	2/3 (66.7%)	1/6 (16.7%)	0.23 ^b^
Livebirth rate						
ET	34/84 (40.5%)	29/96 (30.2%)	0.16 ^b^	2/6 (33.3%)	0/11 (0%)	0.11 ^b^
FET	29/89 (32.6%)	52/145 (35.9%)	0.67 ^b^	1/6 (16.7%)	5/11 (45.5%)	0.33 ^b^

*p*-value was calculated by ^a^ Chi-squared test or ^b^ Fisher’s exact test.

**Table 7 jpm-13-01079-t007:** Clinical variables associated with rates of clinical pregnancy, miscarriage and cumulative pregnancy by multivariate logistic regression analysis.

Variables	Clinical Pregnancy Rate		Miscarriage Rate		Cumulative Pregnancy Rate	
	Odds Ratio (95% CI)	*p*-Value	Odds Ratio (95% CI)	*p*-Value	Odds Ratio (95% CI)	*p*-Value
Male age	0.97 (0.92–1.03)	0.36	0.90 (0.82–0.98)	0.01	1.03 (0.98–1.10)	0.24
Female age	0.90 (0.83–0.98)	0.01	1.08 (0.95–1.22)	0.24	0.86 (0.80–0.94)	<0.0001
AMH	0.99 (0.97–1.01)	0.40	1.01 (0.98–1.05)	0.37	0.99 (0.98–1.01)	0.34
Female BMI	1.02 (0.97–1.06)	0.47	1.16 (1.06–1.27)	0.001	0.99 (0.96–1.04)	0.88
Male BMI	0.99 (0.98–1.00)	0.09	1.01 (1.00–1.03)	0.05	0.99 (0.98–1.00)	0.07
Estradiol level at trigger	1.00 (1.00–1.00)	0.08	1.00 (1.00–1.00)	0.50	1.00 (1.00–1.00)	0.43
Gonadotrophin total dosage administered	1.00 (1.00–1.00)	0.37	1.00 (1.00–1.00)	0.31	1.00 (1.00–1.00)	0.12
SDF	1.00 (0.98–1.02)	0.96	0.97 (0.92–1.02)	0.21	1.03 (1.00–1.07)	0.06
Duration of infertility	0.98 (0.88–1.09)	0.72	0.89 (0.71–1.10)	0.28	0.98 (0.87–1.10)	0.72
Total mature oocytes	1.09 (1.01–1.17)	0.02	1.03 (0.75–1.41)	0.88	0.89 (0.77–1.02)	0.09
Type of ejaculation	1.03 (0.68–1.58)	0.88	1.36 (0.64–2.90)	0.43	0.40 (0.11–1.42)	0.15
Sperm preparation method	0.86 (0.74–0.99)	0.04	0.39 (0.14–1.04)	0.06	1.14 (0.72–1.81)	0.58

## Data Availability

Under the terms for accreditation, the data in this study are not placed in a publicly accessible repository but can be sourced by correspondence with the Medical Director; corresponding author.

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
