# Peer review of "The Sperm DNA Fragmentation Assay with SDF Level Less Than 15% Provides a Useful Prediction for Clinical Pregnancy and Live Birth for Women Aged under 40 Years"

_jpm, 2023, doi:10.3390/jpm13071079_

Round 1
Reviewer 1 Report
As a reviewer, I would like to provide the following comments regarding the retrospective cohort study:
1. The study design and population are well-defined, focusing on males who underwent a sperm DNA fragmentation (SDF) assay and participated in various assisted reproductive technology (ART) cycles at the PIVET Medical Centre.
2. It is interesting to note that unadjusted fertilization rates did not differ between the SDF groups, while the adjusted fertilization rates revealed a significant difference among women younger than 35 years with SDF levels <15%. This highlights the importance of considering oocyte quality when analyzing fertilization rates.
3. The finding that there was no overall difference in blastocyst development and clinical pregnancy rate in women between the SDF groups, expect for a higher pregnancy rate in women younger than 35 years with lower SDF levels, is intriguing. It might be useful to explore potential contributing factors or confounders that could explain this observation.
4. The lack of an overall difference in miscarriage and live birth rates between the two SDF groups suggests that SDF levels may not be the sole determinant of ART success. Further research is needed to understand the complex interplay between SDF levels and other factors that may influence ART outcomes.
5. The study found the highest cumulative live birth rate among the lower SDF group with the IVF mode, particularly in women aged <40 years. The fact that the IVF case numbers were disproportionately low could be a limitation in a study, and it would be important to mention this in the discussion section.
6. It would be helpful to provide a more detailed description of the study population, including demographic and clinical characteristics such as male and female age, BMI, duration of infertility, and lifestyle factors. This will help readers better understand the context of the results and identify potential confounders.
7. The discussion section could benefit from a more thorough comparison of the study findings with current literature, as well as a discussion of the clinical implications of the results, limitations of the study, and potential directions for future research.
Overall, the quality of English is quite good, with clear and concise language used to describe the study and its findings. However, there are a few areas where improvements can be made to enhance clarity and readability:
1. Use consistent terminology throughout the text. For example, when referring to the two SDF groups, maintain the same format (either ‘’<15%’’ and ‘’³15%’’ or ‘’lower’’ and ‘’higher’’) for consistency.
2. Ensure that all abbreviations are defined upon first use.
3. Some sentences are quite long and might be easier to understood if broken into shorter, more straightforward sentences.
4. In a few instances, there are minor grammatical errors or awkward phrasing. For example, ‘’there was a highly significant fertilization rate in favour of SDF<15%’’ could be rephrased as ‘’there was a highly significant difference in fertilization rates in favour of the group with SDF levels <15%’’.
By addressing these minor issues and polishing the language, the text will be even clearer and more effective in conveying the study’s findings.
Author Response
Reviewer #1, response to itemised comments
- No action required
- Favourable comments, no action required
- Questions concerning the overall blastocyst development rates and clinical outcomes being similar in the group of SDF <15% and SDF ≥15% but it revealed with a significant higher livebirth rate in SDF <15% in women aged <35 years. There were no overall significance differences in the laboratory (except the adjusted fertilization rate) and clinical outcomes. However, when stratifying into women’s age groups, the clinical pregnancy and livebirth rate were significantly higher in SDF <15% for women aged <35 years. We have added data analysed by logistic regression to ascertain possible confounding factors - this is shown in Table 7.
- The reviewer has rightly pointed this out. Relevant points have been added in the discussion.
- The low number of cases in the IVF group has been elaborated in the discussion.
- Additional variables such as infertility duration, gonadotrophin total dosage administered, Estradiol level at trigger, day of trigger, day of OPU, number of oocytes retrieved and mature oocytes were added.
- The discussion has been revised as suggested.
The comments on the Quality of English Language noted and revised accordingly.
Reviewer 2 Report
Dear Authors,
this is an elegant study on sperm DNA fragmentation or DNA fragmentation index impact on laboratory and clinical success in terms of pregnancy rate in infertile couples.
Your introduction sufficiently and precisely pointed the goal based on current knowledge.
Methodology and analysis were consistent and comprehensive.
Results despite vast amount od data were presented clearly and easy-to-read.
Discussion was constructed in a logic and critical way with adequate references.
Conclusions are drawn carefully and adequately to established facts and are clinically very useful.
Congratulations.
Single word error found in line 268 (should be 2.4.3. Sample Washing).
Author Response
Most comments favourable; no action required.
Single comment on line 268 corrected:
Simple sperm washing; This is terminology consistent with WHO; ref 37
Round 2
Reviewer 1 Report
I am pleased to accept the article for publication in its current form.